chemical physics/synthetic chemistry

hydrophobically associating polymer, aggregation behaviour, hydrophobicity microzone, resistance coefficient, residual resistance factor, porous media

**Author for correspondence:**
Shijie Zhu
e-mail: 289045557@qq.com

This article has been edited by the Royal Society of Chemistry, including the commissioning, peer review process and editorial aspects up to the point of acceptance.

# The seepage flow characteristics of hydrophobically associated polymers with different aggregation behaviours in porous media

Leiting Shi[1], Shijie Zhu[1], Zhongbin Ye[1], Jian Zhang[2],
Xinsheng Xue[2] and Wensen Zhao[2]

[1]State Key Laboratory of Oil and Gas Reservoir and Exploitation Engineering, Southwest Petroleum University, Si Chuan, Chengdu 610500, People's Republic of China
[2]State Key Laboratory of Offshore Oil Exploitation, Beijing 100027, People's Republic of China

SZ, 0000-0002-2193-2646

The polymer solution for oil displacement is subjected to strong shear action in practical application, and this action will affect its percolation characteristics in porous media. The effects of mechanical shearing on the solution properties and seepage characteristics of modified hydrophobically associated polymers and dendrimers with two different aggregation behaviours were studied. The results showed that mechanical shearing did not affect hydrophobic microzones. Polymers can re-associate to restore part of the network structure, thereby improving shear resistance (dendritic hydrophobically associating polymers > hydrophobically modified partially hydrolysed polyacrylamide). Polymers with 'cluster' aggregation behaviour enhanced solution performance, enabling them to establish higher resistance coefficient (RF) and residual resistance factor (RRF) in porous media but also bringing about injection difficulties. Increasing the injection rate would increase the injection pressure, but the established RF and RRF showed a downward trend. Mechanical shear pretreatment effectively improved the injectability of the polymer. To achieve polymer injection and flow control, pre-shearing polymer solution and low-speed injection can be used in field applications.

# 1. Introduction

The mechanism of action of polymers is to increase the solution viscosity of the displacement phase and reduce the permeability of the displacement phase to achieve the fluidity control effect, thereby improving the displacement efficiency and sweep efficiency [1,2]. Thus, polymer solution is widely used and is one of the main methods used to improve oil recovery in oilfields. The use of polymers to control the stability and flocculation behaviour of some dispersions has great technological importance [3] (table 1).

However, the process of polymer injection into the formation induces strong shear action (for example, the polymer injection pump, borehole, perforation hole, etc.) [4,5]. Mechanical shear will inevitably occur in the process of polymer flooding, which will lead to a significant reduction in the performance of polymer solutions and subsequently affect polymer flooding [6]. Therefore, improving the shear resistance of polymer solutions for oil displacement is the main direction of scientific researchers at present. Among them, hydrophobically associated polymer is a typical polymer flooding agent with better shear resistance [7–11]. Given its unique association, polymer molecular chains can form a stronger three-dimensional structure, making the apparent viscosity of polymer solution change from a single bulk viscosity to a comprehensive presentation of bulk viscosity and structural viscosity. Especially after the critical association concentration, the structural viscosity of the solution increases greatly, and the apparent viscosity of the solution increases greatly. After mechanical shearing, the molecular chain of the polymer is broken, the solution structure is damaged and the solution viscosity is reduced. Although bulk viscosity is irreversibly destroyed by mechanical shear, hydrophobic association can restore part of the structural viscosity, thus restoring the solution viscosity.

Hydrophobically associated polymers exhibit good shear resistance mainly due to the restoration of space network structure and structural viscosity under association. The size of association depends on the spatial distribution of hydrophobic groups in polymer solutions. Therefore, the influence of polymer aggregation behaviour on its shear resistance is particularly important [12,13]. Based on the general linear hydrophobic associative polymers, Francois' [14] team used the topological structure theory to divide hydrophobic associative polymers into two structural models: grafted and far-out polymers (figure 1). The former includes the random and continuous block distribution of hydrophobic groups in the main chain and the end of hydrophilic barrier. Hydrophobic modified polyacrylamide polymer (HMPAM) is a typical representative of this type of polymer [15–17]. The latter mainly includes 'star' polymers with hydrophobic groups attached to one or both ends of the hydrophilic main chain and one hydrophobic group connected to multiple hydrophilic main chains [18–20]. Research demonstrates that the spatial structure of the polymer molecule is strengthened, the distribution of hydrophobic groups is more homogeneous after shearing, and the association effect can be better exerted such that the polymer molecule exhibits stronger shearing resistance.

To further improve the field application of polymer flooding, it is necessary to recognize the percolation characteristics of hydrophobically associated polymers in porous media after mechanical shearing. It is necessary to study the porous media percolation characteristics of hydrophobic associating polymers with two different molecular structure models by considering the actual mechanical shear effect in the field and the mechanical degradation in the application of polymer flooding. One is a hydrophobically modified partially hydrolysed polyacrylamide, and the other is a branched hydrophobically associated polymer. The main research contents include three aspects: (i) the aggregation behaviour characteristics of two types of polymers; (ii) the influence of shearing on the viscosity enhancement, aggregation behaviour, microstructure and static adsorption properties of polymer solutions; and (iii) the percolation characteristics of polymers before and after mechanical shearing in porous media at different velocities.

# 2. Experiment

## 2.1. Materials and equipment

Chemicals: sodium chloride, calcium chloride, magnesium chloride, acrylamide (AM), acrylic acid (AA), two methyl allyl pair of 16 alkyl benzyl ammonium chloride, two methyl allyl-N-alkyl ammonium chloride, maleic anhydride, macromolecular skeleton monomer. Chemicals are analytical reagent (AR).

Experimental brine: The experimental brine with 3000.00 mg l$^{-1}$ Na$^+$ and 300.00 mg l$^{-1}$ Ca$^{2+}$/Mg$^{2+}$.

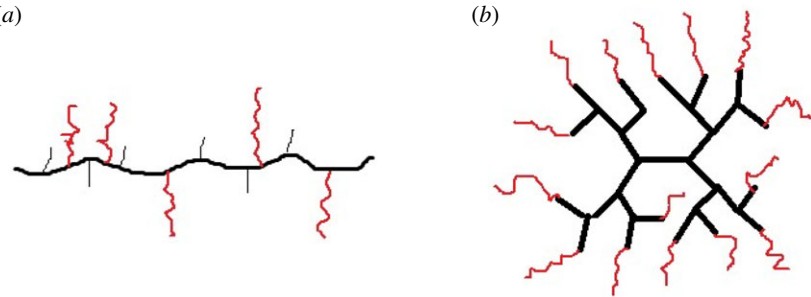

**Figure 1.** Molecular characteristic diagram. (*a*) Grafted polymer and (*b*) far-out polymer.

**Table 1.** Nomenclature.

| | |
|---|---|
| RF | resistance coefficient |
| RRF | residual resistance factor |
| $\eta_r$ | relative viscosity, dimensionless |
| $\eta_{sp}$ | specific viscosity, dimensionless |
| $\eta$ | viscosity of the sample solution, mPa s |
| $\eta_0$ | viscosity of blank solvents, mPa s |
| $\Gamma$ | adsorption quantity, $\mu$g g$^{-1}$ |
| $V$ | volume of polymer solution, ml |
| $C_0$ | initial concentration of polymer solution, mg l$^{-1}$ |
| $C_e$ | concentration of polymer solution after adsorption equilibrium, mg l$^{-1}$ |
| $G$ | quality of quartz sand, g |
| $R$ | dynamic retention of polymer in core, $\mu$g g$^{-1}$ |
| $C_{01}$ | the inlet concentration of polymer solution, mg ml$^{-1}$ |
| $C_i$ | the outlet concentration of polymer solution, mg ml$^{-1}$ |
| $V_0$ | the injection volume of polymer solution, ml |
| $V_i$ | the output volume of polymer solution, ml |
| $W$ | the dry weight of the core, g |

Instruments: (i) fluorescence spectrometer (Cary Eclipse); (ii) the Ubbelohde viscometer; (iii) scanning electron microscope (SEM); (iv) displacement experimental equipment; (v) ultraviolet spectrophotometer; (vi) nanoscope IIIa microscope.

Two types of hydrophobically associated polymers [15] with different molecular structures have been studied: one is HMPAM with a hydrophobic group content of 1.2 mol% and molecular weight of 12 million; the other is dendritic hydrophobically associating polymers (DHAP) with a hydrophobic group content of 0.6 mol% and molecular weight of 6 million.

(1) Synthesis of hydrophobically modified partially hydrolysed polyacrylamide. The molecular structure is presented in figure 2. The synthetic steps are as follows. The first step involves weighing 12 g of AM, 3 g of AA, 0.369 g of two methyl allyl pair of 16 alkyl benzyl ammonium chloride and 35 g of distilled water to a beaker, stirring and dissolving evenly. Then, the pH is adjusted to 5–6 by using sodium carbonate. After heating for 45 min in a water bath at a temperature of 45°C, adding 4.8 mg initiator sodium bisulfite and 4.8 mg ammonium persulfate, and reacting for 6–10 h, the hydrophobically modified partially hydrolysed polyacrylamide polymer could be obtained.

(2) Synthesis of branched hydrophobically associating polymers. The molecular structure is presented in figure 3. Synthesis steps are as follows. First, the two generations of the polyamide amine tree, including 2.619 g macromolecular skeleton monomer and 1.437 g maleic anhydride, were added to different beakers. Then, 20.16 and 8.09 g of two methylene sulfoxide are added, and the

**Figure 2.** The molecular structure of HMPAM.

**Figure 3.** The molecular structure of DHAP.

solution is evenly stirred. Using a molar ratio of 1 : 8, the two methylene sulfoxide solutions of skeleton monomer are slowly dripped into the two methylene sulfoxide solutions of maleic anhydride, and the nitrogen is added via a dropping process. The system is placed into an ice salt bath. After adding the skeleton monomer solution, the solution is stirred until the temperature of the reaction liquid system no longer increases (temperature is less than 50°C). The functional skeleton monomer ($C_{94}H_{144}O_{36}N_{26}$) is obtained by repeated precipitation with chloroform. Then, 20 g AM, 5 g AA, 0.025 g functionalized skeleton monomer, 0.8 g two methyl allyl -N- alkyl ammonium chloride and 74 g distilled water are weighed. The components are placed into the beaker to stir and dissolve. Sodium hydroxide is used to adjust pH to 5–6, and the solution is incubated at 35°C to preheat for 30 min. Then, 0.26 g solvent urea, 0.26 g sodium sulfate, 0.052 g chain transfer agent sodium methoxide, 4.8 mg oxidation–reduction initiator sodium bisulfite and 4.8 mg ammonium persulfate are added. The reaction proceeds for 8 h, yielding DHAP polymer.

## 2.2. Method

### 2.2.1. Simulation of mechanical shear conditions

The polymer solution (200 ml) was injected into the Waring blender and sheared by the Waring blender at different speeds.

### 2.2.2. Determination of apparent viscosity and intrinsic viscosity

Polymer solution viscosities before and after shearing using a Waring blender were measured using a Brookfield DV-III viscometer with a shearing rate of $7.34 \, s^{-1}$ at 25°C [21]. Then, the curve of the viscosity–concentration relationship was plotted to calculate the retention rate of viscosity.

### 2.2.3. Determination of intrinsic viscosity

To reflect the change of molecular weights through the shearing effect of the Waring blender on HMPAM and DHAP, the change of intrinsic viscosity was measured using the Ubbelohde viscometer. The Ubbelohde viscometer was cleaned with water many times until there was no residual polymer on the inner surface of glass and was then sterilized. The concentration range of the prepared polymer is 500, 400, 300, 200, 100 and $0 \, mg \, l^{-1}$ (blank solvents). Polymer solutions were injected into the Ubbelohde viscometer, and the flow times through the ball were recorded in parallel three times until flowing time difference was less than 0.2 s. The average value of three times was used as the final data.

Characteristic viscosity [22,23]: According to the Huggins equation, the relationship between $\eta_{sp}/C$ and polymer concentration $C$ was drawn by measuring the viscosity of polymer solution with different concentrations, taking polymer concentration $C$ as the abscissa and $\eta_{sp}/C$ as the ordinate. By extrapolating the line of $\eta_{sp}/C$ to zero concentration, the value corresponding to the intersection point

**Table 2.** Standard curve formula of polymer absorption. $A$ is UV absorbance, AU; $c$ is concentration, mg l$^{-1}$.

| polymer | relational expression | degree of correlation, $R^2$ |
|---|---|---|
| HMPAM | $A = 0.0077c + 0.0800$ | 0.9937 |
| DHAP | $A = 0.0109c + 0.1675$ | 0.9955 |

is equal to the intrinsic viscosity $\eta$, and the Huggins constant $K_H$ can be obtained by slope. For the calculation of the relative viscosity $\eta_r$ and specific viscosity $\eta_{sp}$, see the below equations

$$\eta_r = \frac{\eta}{\eta_0} \tag{2.1}$$

and

$$\eta_{sp} = \eta_r - 1. \tag{2.2}$$

### 2.2.4. Fluorescence analysis of polymer aggregation behaviour

Using a fluorescence photometer (Cary Eclipse) with the help of fluorescence probe molecule [24–26], the sensitivity of microphysical environment to photophysical and photochemical properties is studied. The concrete steps are as follows: (i) prepare pyrene probe and prepare a solution of $1.715 \times 10^{-6}$ mol l$^{-1}$ pyrene; (ii) select the polymer solution required for the preparation of the target; (iii) place 0.5 ml containing pyrene into the ampoule containing the target polymer solution and set it up for 24 h; (iv) administer the fluorescence photometer test of the radical association peak. When the ground-state pyrene in the solution is close enough to the excited state pyrene (about 4 Å) and forms a surface-to-surface lamination conformation, a ground-state pyrene molecule can combine with it to form excimer before the decay of excited state pyrene molecule

$$Py^* + Py \rightarrow (Py\,Py)^*. \tag{2.3}$$

$Py^*$ is an excited state pyrene, Py is a ground-state pyrene and $(Py\,Py)^*$ is an excimer.

The fluorescence peak of $(Py\,Py)^*$ appears in the long-wave region of the fluorescence emission spectrum, called the excipient association peak.

### 2.2.5. Analysis of polymer micromorphology

Micromorphology is analysed using an environmental scanning electron microscope [27,28]. The concrete steps are as follows: (i) configure a polymer solution into a target concentration solution; (ii) place a drop of polymer solution on the metal sample rack and fix it, use liquid nitrogen to solidify the solution quickly and lock the structure form in the solution, and then, sublimate the water molecule in the freeze vacuum dryer; (iii) a special device is used to cover the surface of the sample; (iv) observe the microstructure of the polymer solution using a scanning electron microscope.

### 2.2.6. Static adsorption measurement

The concentration change of polymer solution was analysed by ultraviolet spectrophotometer to assess static adsorption and dynamic retention. Establishment of standard curve of concentration and absorbance: two types of polymer solutions with concentrations of 20, 40, 60, 80, 100, 120, 140 and 160 mg l$^{-1}$ were prepared. The standard curve of the relationship between the absorbance of polymer solution and polymer concentration was established. See table 2.

Static adsorption [29–31]: The immersion method of polymer was mainly studied using the immersion method. A ratio of polymer to sand of $10:1$ was immersed for 24 h. The supernatant was separated by centrifugation, and the absorbance of polymer in solution was determined. The static adsorption capacity was calculated using the following equation:

$$\Gamma = \frac{V(C_0 - C_e)}{G}. \tag{2.4}$$

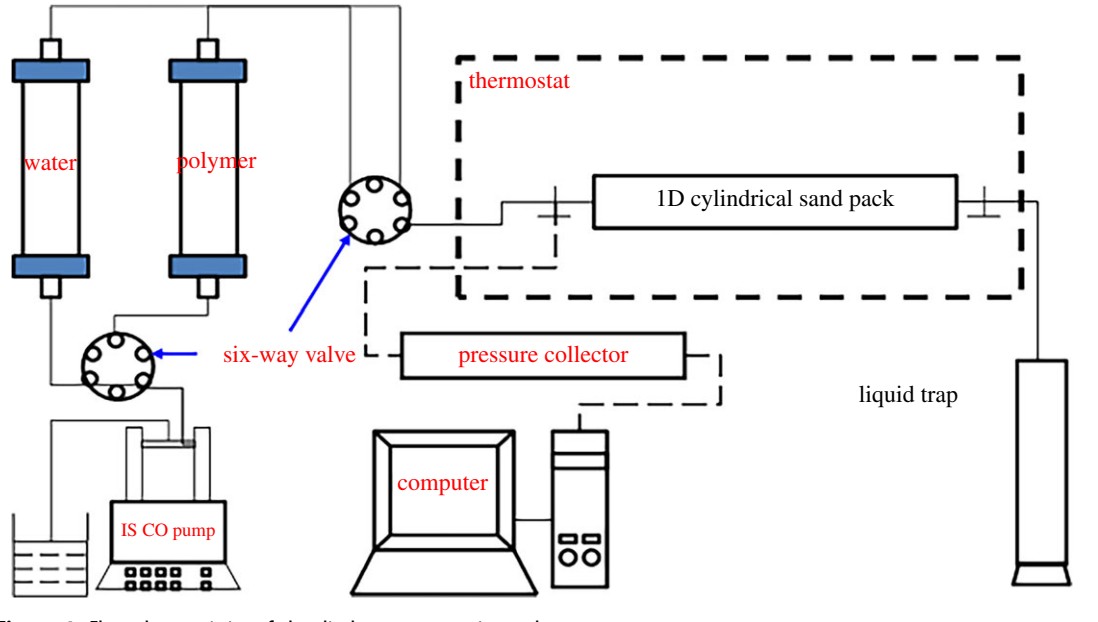

**Figure 4.** Flow characteristics of the displacement experimental process.

### 2.2.7. Dynamic retention measurement

Dynamic retention [32–34]: Based on the percolation characteristic experiment described in §2.2.8, the concentration and volume of polymer at the end of core outlet are continuously measured. Using the principle of material balance, the concentration of injection minus the concentration of output is used to calculate the retention of polymer in core as follows:

$$R = \frac{C_{01}V_o - \sum_{i=1}^{n} C_i V_i}{W}. \tag{2.5}$$

### 2.2.8. Experimental study on percolation characteristics of polymers in porous media

(1) The experimental contents are as follows: two polymer solutions with concentration of 2000 mg l$^{-1}$ were prepared. A part of the solution was used to simulate mechanical shear. The simulation condition was 3500 r.p.m. shear for 20 s. The prepared polymer solution was injected into porous media at a certain injection rate of 0.5, 1 and 3 ml min$^{-1}$, respectively.

   (2) The research adopts displacement experimental process equipment as described in figure 4. The concrete steps are as follows [35–37]: (i) The sand pack was vacuumed and saturated with the synthetic brine. Porosity was calculated based on the weight difference before and after the sand pack was saturated with brine. Absolute permeability was obtained by injecting brine at three different flow rates and calculated according to Darcy's Law. The permeability is approximately $2000 \times 10^{-3}$ µm$_r^2$ and the porosity is approximately 30%. (ii) When synthetic brine was injected at a rate of 3.0 ml min$^{-1}$, the stabilized pressure difference across the sand pack was recorded and denoted as $\Delta P_b$. (iii) Polymer solution was injected at the set speed, and the stable pressure difference was recorded as $\Delta P_p$. Afterward, the brine injection was switched on. The stable pressure difference of the brine injection at this stage was denoted as $\Delta P_a$. (iv) Pressure values were recorded at two locations as shown in figure 5 for the entire experimental process. (v) According to the definitions of RF and RRF, these parameters were calculated as follows:

$$RF = \frac{\Delta P_p}{\Delta P_b} \tag{2.6}$$

and

$$RRF = \frac{\Delta P_a}{\Delta P_b}. \tag{2.7}$$

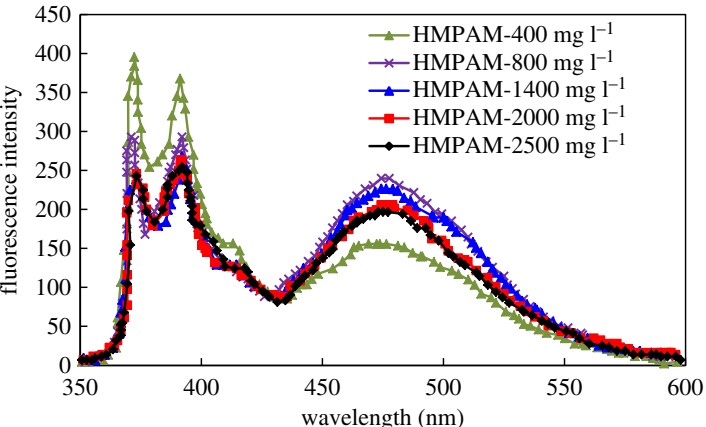

**Figure 5.** Fluorescence spectra of HMPAM.

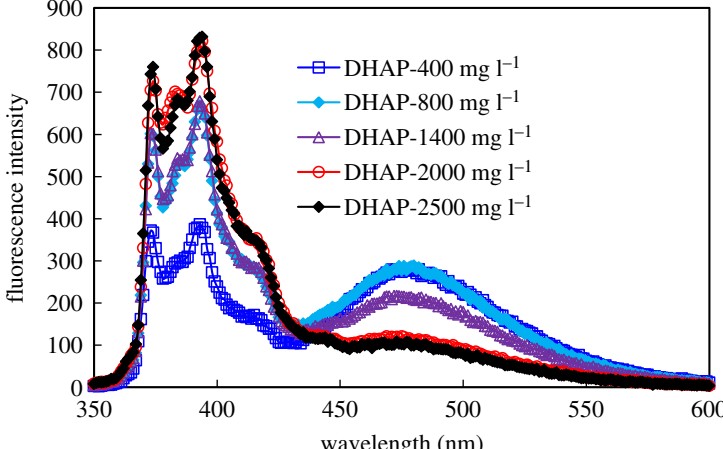

**Figure 6.** Fluorescence spectra of DHAP.

# 3. Results and discussion

## 3.1. Analysis of aggregation behaviour of polymers

The polymers used in experiments are hydrophobically associated polymers. Pyrene is a non-polar substance. It is almost insoluble in polar solvents, but it can be solubilized into non-polar hydrophobic microdomains. It is an ideal probe to characterize the polar environment. Therefore, when studying the aggregation behaviour of two hydrophobically associated polymer molecules in the solution, pyrene can be used as a probe to study the internal microstructure of the solution.

Figures 5 and 6 present the fluorescence spectra of HMPAM and DHAP at different concentrations. The original data are provided in the electronic supplementary material. The chart presents the location of various emission peaks. The first emission peak $I_1$ and the third emission peak $I_3$ appear near 375 and 385 nm, respectively, and the base excitation association peak $I_e$ appears near 475 nm.

Figures 5 and 6 reveal that when the polymer concentration increases, the fluorescence intensity differences that correspond to the peaks are obvious in two types of hydrophobic-associated polymer fluorescence spectra. This finding demonstrates that as the concentration increases, the aggregation behaviour of hydrophobically associated polymer in solution differs significantly.

The ratio of the intensity of the third emission peak $I_3$ to the first emission peak $I_1$ is often used to characterize the polar environment of the solution. Figure 7 is the curve of the concentration variation of the $I_3/I_1$ ratio of the two solutions.

The $I_3/I_1$ ratio indicates that the two types of polymers increase first and then become stable as the concentration increases. The difference is that after the concentration of HMPAM is greater than 400 mg l$^{-1}$, the increasing speed of $I_3/I_1$ is quicker, and it tends to stabilize at a concentration of

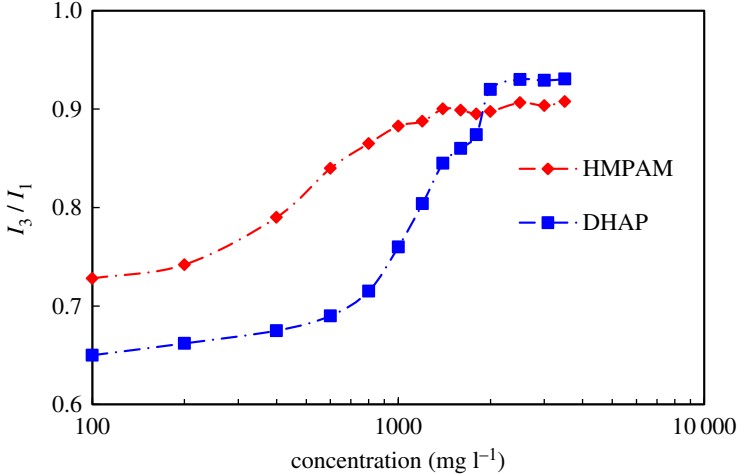

**Figure 7.** The relationship between the $I_3/I_1$ ratio and the concentration of polymer solution.

1600 mg l$^{-1}$. The $I_3/I_1$ value of DHAP solution increases rapidly to 800 mg l$^{-1}$ and is stable after the concentration is greater than 2000 mg l$^{-1}$. This finding is explained by the fact that it is difficult for the branched structure of DHAP to make contact with each other. The hydrophobic base contacts with the branched chains are more likely to exhibit intramolecular associations; thus, there are less hydrophobic microdomains than HMPAM. As the concentration of the polymer increases, the intermolecular association occurs in both types of polymers. In addition, due to the molecular configuration characteristics of DHAP, the first small number of intermolecular small aggregates should be formed rather than local or even overall network structure [23,24]. Therefore, $I_3/I_1$ will increase slowly. When the polymer concentration exceeds the critical association concentration, the intermolecular association starts to dominate, and the hydrophobic microarea is generated. The increasing trend of $I_3/I_1$ is transformed and stabilized. The network structure of the solution is formed in the solution.

As the polymer concentration increases, the hydrophobic microzone formed by the hydrophobically associating polymer (HMPAM and DHAP) leads to a change in the polarity of the solution. The characteristics of different curves show that the aggregation behaviour of the two polymers is different in aqueous solution.

## 3.2. Effect of shear action on polymer

When polymer solution is injected into the porous media, it undergoes strong shear action. Therefore, when studying polymer mobility in porous media, the effect of shearing on the solution is mainly considered.

### 3.2.1. Effect of mechanical shearing on viscosity

The effect of shear strength on the intrinsic viscosity and Huggins constant of HMPAM and DHAP are presented in table 3.

The apparent viscosities of both polymers decrease rapidly as shear strength increases, and the apparent viscosities of polymer DHAP are relatively high, which is also due to its strong spatial network structure that enhances its structural viscosities. In addition, as shear strength increases, the intrinsic viscosity of HMPAM decreases more. Compared with DHAP (molecular weight 6 million), HMPAM is a flexible linear polymer with a higher molecular weight (12 million), and its molecular conformation in aqueous solution is more relaxed. When subjected to shear stress, the molecular chain is more likely to break, leading to a faster decrease in intrinsic viscosity. DHAP is a branched polymer with lower molecular weight, stronger rigidity of molecular chains, greater intramolecular associations and closer interactions among branched chains. Thus, it exhibits better resistance to external shear stress. When subjected to shear stress, the intrinsic viscosity decreases slowly.

As a criterion to measure the hydrophobic effect, the Huggins constant $K_h$ reflects the interaction strength between macromolecules or molecular chains and solvent molecules. Compared with the data presented in table 3, we can see that (i) $K_h$ constants of the two polymers decrease as shear

**Table 3.** The effect of shear strength on the intrinsic viscosity and Huggins constant of polymer.

| polymer | shear strength | apparent viscosity, mPa s | intrinsic viscosity, ml g$^{-1}$ | Huggins constant, $K_H$ |
|---|---|---|---|---|
| HMPAM | no shear | 166.25 | 1741.82 | 1.74 |
| | 20 s at 1 speed | 134.10 | 1445.71 | 1.35 |
| | 20 s at 2 speeds | 99.81 | 1127.65 | 0.98 |
| | 20 s at 3 speeds | 45.36 | 836.07 | 0.87 |
| DHAP | no shear | 210.23 | 812.56 | 7.85 |
| | 20 s at 1 speed | 187.25 | 793.82 | 7.63 |
| | 20 s at 2 speeds | 157.35 | 747.56 | 5.48 |
| | 20 s at 3 speeds | 92.41 | 650.05 | 4.12 |

**Table 4.** Effect of different shear strengths on the number of hydrophobic microregions in polymer solution.

| polymer | shear mode | $I_e/I_m$[a] | polymer | shear mode | $I_e/I_m$ |
|---|---|---|---|---|---|
| DHAP -2000 mg l$^{-1}$ | no shear | 0.21 | HMPAM -2000 mg l$^{-1}$ | no shear | 0.81 |
| | 20 s at 1 speed | 0.22 | | 20 s at 1 speed | 0.83 |
| | 20 s at 2 speeds | 0.20 | | 20 s at 2 speeds | 0.86 |
| | 20 s at 3 speeds | 0.22 | | 20 s at 3 speeds | 0.88 |

[a]$I_e$ is the maximum excipient association peak; $I_m$ is the maximum emission peak of monomer in emission spectrum.

strength increases, and the shear effect will affect the hydrophobic effect to some extent; and (ii) the DHAP interaction is stronger than that of HMPAM, but the hydrophobic group content of DHAP is lower than that of HMPAM (see Materials and equipment), indicating that the hydrophobic association has a better effect on dendrimers with stronger spatial structure.

### 3.2.2. Effect of mechanical shearing on the hydrophobic microzone

The experiment assesses the change in the number of hydrophobic microregions as shear strength increases in a solution with 2000 mg l$^{-1}$ polymer. The results are shown in table 4.

As shear stress increases, the $I_e/I_m$ of 2000 mg l$^{-1}$ HMPAM solution gradually increases albeit modestly. The shear stress destroys the hydrophobic effect between polymer molecules and aggregates to a certain extent, reducing the number of hydrophobic regions. For 2000 mg l$^{-1}$ DHAP solution, the $I_e/I_m$ does not change as the shear effect increases, indicating that shear effect has minimal effects on the number of hydrophobic microdomains, which do not affect the hydrophobic groups of polymers, and polymer molecules could be relinked through hydrophobic groups.

### 3.2.3. Effect of mechanical shearing on microstructure

Two types of polymer solutions (DHAP and HMPAM) at concentrations of 2000 mg l$^{-1}$ are prepared using experimental brine. The micromorphology of the polymer before and after shearing are analysed by the scanning electron microscope. Among them, the micromorphology of the branched polymer DHAP is shown in figure 8. The micromorphology of the chain polymer HMPAM is shown in figure 9.

(i) When the polymer concentration is 2000 mg l$^{-1}$, the microchain bundle formed by the polymer DHAP is more robust than that of HMPAM; (ii) shearing action will fracture the polymer intermolecular packing. Based on the effect of shear action on the hydrophobic microregion, we think the shear mainly destroys the polymer molecular intermolecular packing. The hydrophobic region enables fractured intermolecular packing reassembly, so they can still form the space network structure [24–26]. Thus, the viscosity concentration curve of polymer reveals shear thickening ability, shows stronger spatial structure and is less affected by the shear effect. Overall, the shear affects the molecular structure of chain hydrophobically associated polymer to an extent, but the intermolecular

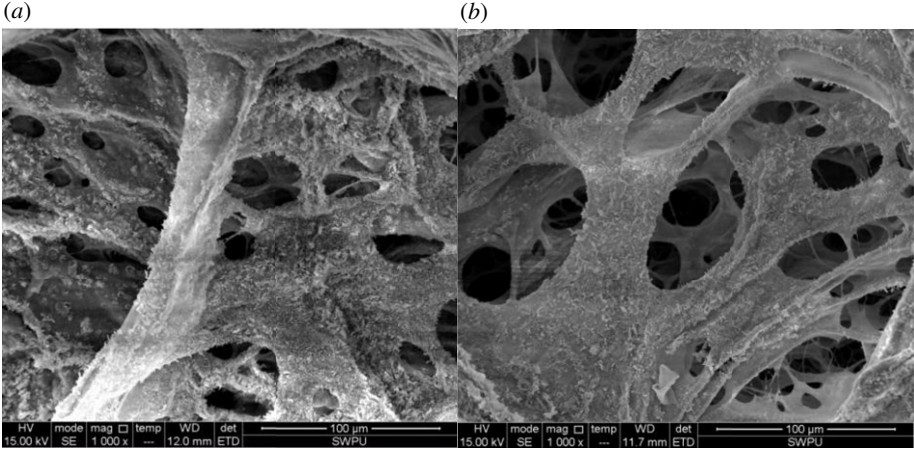

**Figure 8.** Micromorphology of polymer DHAP. (*a*) Before shearing and (*b*) after shearing.

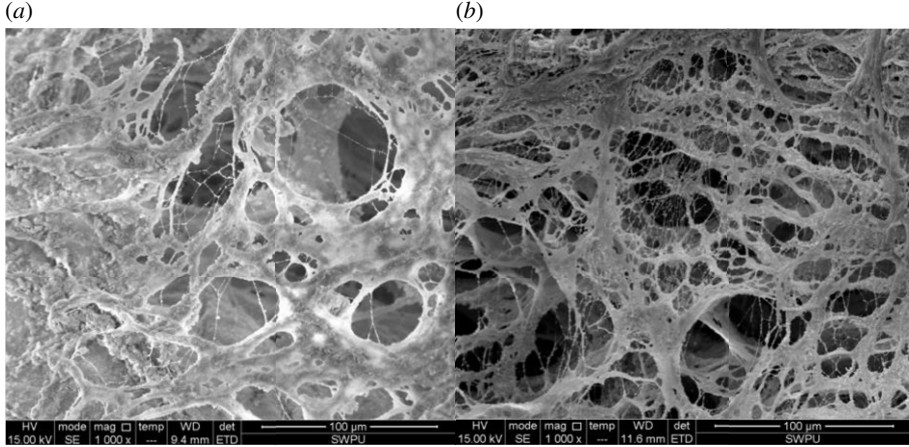

**Figure 9.** Micromorphology of polymer HMPAM. (*a*) Before shearing and (*b*) after shearing.

packing cleavage in the hydrophobic group under the effect of association ensures that the growth trend of solution viscosity and the molecular structure also disrupt viscosity influence.

### 3.2.4. Effect of mechanical shearing on static adsorption

The static adsorption characteristics of two polymers before and after mechanical shearing are presented in figure 10.

The adsorption characteristics of hydrophobically associating polymers on quartz sand surface can be divided into three stages. The first stage is the stage of low adsorption concentration, in which intramolecular association mainly occurs, and surface adsorption occurs depending on the interaction force between polymer molecular clusters and the rock surface. Compared with unsheared polymer solution, the hydrodynamic size of sheared polymer aggregates is small. When adsorbed on the surface of quartz sand, the steric hindrance effect between them is weakened. Thus, the adsorption density is higher, which leads to larger adsorption capacity. In the second stage, the adsorption capacity increases rapidly as the solution concentration increases. The proportion of intermolecular association increases gradually. Polymer molecules originally adsorbed on the surface of quartz sand can associate with free polymer molecules in solution, thus forming bilayer or multilayer adsorption on the surface of quartz sand and resulting in a sharp increase in adsorption capacity.

The shear action causes the breakage of some polymer chains and aggregates, weakening the interaction between free polymer molecules and aggregates in solution and polymer molecules on the surface of quartz sand adsorption layer. This effect limits the increase in the thickness of the multi-molecular adsorption layer and leads to a lower rate of increase in adsorption capacity as the polymer concentration increases. The third stage achieves maximum adsorption capacity and maintains

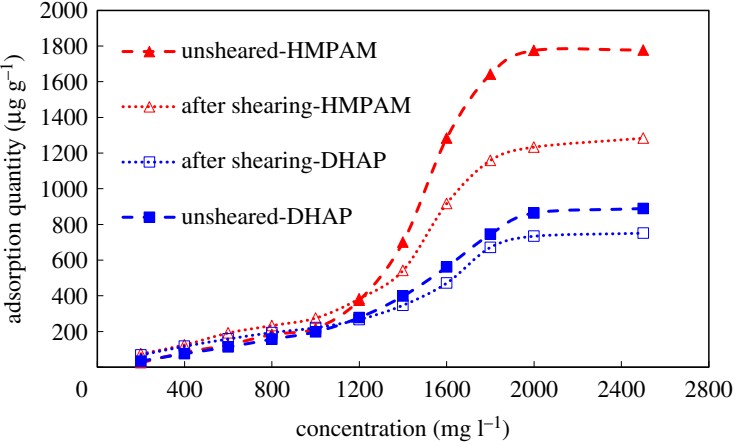

**Figure 10.** Static adsorption characteristics of polymers.

**Table 5.** The dynamic retention of polymers in porous media.

| polymer | shear condition | velocity, ml min$^{-1}$ | permeability, mD | dynamic retention, µg g$^{-1}$ |
|---|---|---|---|---|
| HMPAM | no shear | 0.5 | 2084 | 402.60 |
| | | 1 | 2031 | 209.95 |
| | | 3 | 2079 | 140.50 |
| | 20 s at 1 speed | 1 | 2114 | 160.57 |
| DHAP | no shear | 0.5 | 2034 | 906.64 |
| | | 1 | 2114 | 572.21 |
| | | 3 | 2079 | 313.23 |
| | 20 s at 1 speed | 1 | 2062 | 432.43 |

equilibrium. Among the selected polymers, the equilibrium adsorption capacity (approx. 1700 µg g$^{-1}$) of unsheared HPMA polymer is larger than that of sheared polymer (approx. 1200 µg g$^{-1}$), while that of unsheared polymer (approx. 890 µg g$^{-1}$) is larger than that of sheared polymer (approx. 750 µg g$^{-1}$).

Compared with HMPAM polymer solution, DHAP has a stronger spatial network structure, which makes its intermolecular interaction stronger, resulting in relatively less static adsorption on the surface of quartz sand.

## 3.3. The seepage flow characteristics of polymer in porous media

The effects of mechanical degradation and injection rate on polymer seepage were studied by experiments based on porous media seepage characteristics. The characteristics of adsorption retention, pressure propagation and seepage were analysed.

### 3.3.1. Adsorption retention of polymers in porous media

The effects of injection rate on the dynamic retention of two polymers in porous media and adsorption retention of polymer after shearing for 20 s at 1 speed are shown in table 5.

Table 5 provides three insights: (i) Compared with HMPAM, DHAP has stronger solution properties. On the one hand, the network structure of the solution is maintained, and the retention of the polymer at the pore throat is improved. On the other hand, the polymer molecule in the solution is associated with the polymer molecule adsorbed on the rock surface to form a structure and increased the adsorption capacity, so the retention in the porous medium is relatively large. (ii) As the injection rate increases, the retention of polymer solution of the two structures decreased. The dissolution rate of the reticular structure in the solution was higher than the recombination rate. The size of the molecular aggregates and the retention of the polymer decreased. As the shear action increases, the multilayer structure

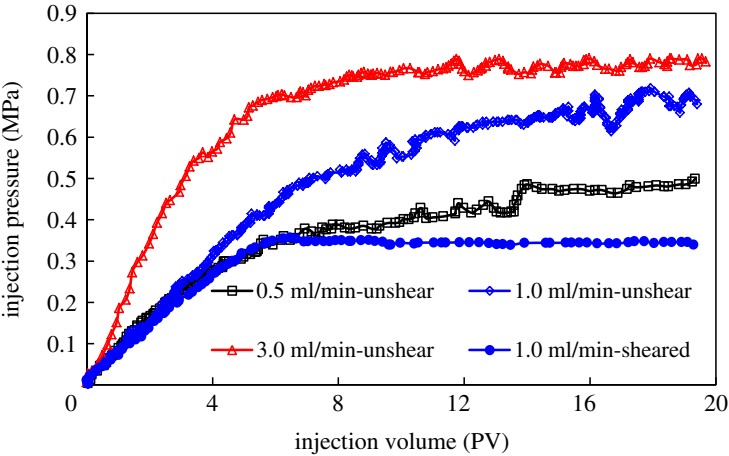

**Figure 11.** Pressure propagation of HMPAM polymer in porous media.

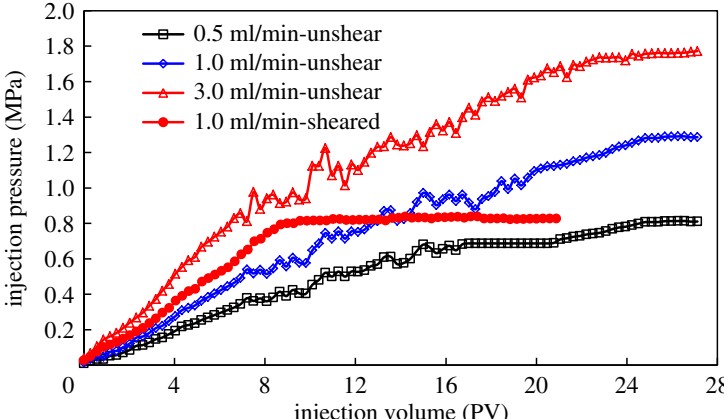

**Figure 12.** Pressure propagation of DHAP polymer in porous media.

adsorbed on the rock surface will also be dismantled, decreasing adsorption capacity. As the injection rate increases, the shear action increases continuously. In addition, the structure of solution formed in porous media is destroyed or dismantled, which reduces the adsorption retention of polymer in porous media. (iii) After shearing, the strength of the network structure of the solution decreases, and the network structure is more easily disassembled. In addition, the adsorption retention decreases.

### 3.3.2. Pressure propagation of polymers in porous media

Pressure propagation characteristics of HMPAM and DHAP polymers in porous media are shown in figures 11 and 12.

There are some differences in injection pressure characteristics between the two polymers in porous media. HMPAM is characterized by two stages of injection pressure, namely, a rapid increase and gradual stabilization. DHAP polymer has three stages: rapidly increasing pressure, pressure increases gradually and final stabilization. During the displacement process, there are obvious pressure fluctuations after accumulative injection of 6PV. On the one hand, the shear action in the displacement process will disintegrate the network structure of polymer solution, and the association can reconstruct the network structure. On the other hand, because the retained polymer at the pore throat is propelled to migrate, pressure fluctuations occur in the dynamic process. The faster the injection rate, the higher the injection pressure, and the more obvious the instability. The stronger the performance of polymer DHAP solution, the higher the injection pressure, and the more obvious the pressure fluctuation.

After mechanical shearing, both polymers show two stages of rapid and stable injection pressure increases without obvious pressure dynamic characteristics. Shear action destroys the polymer

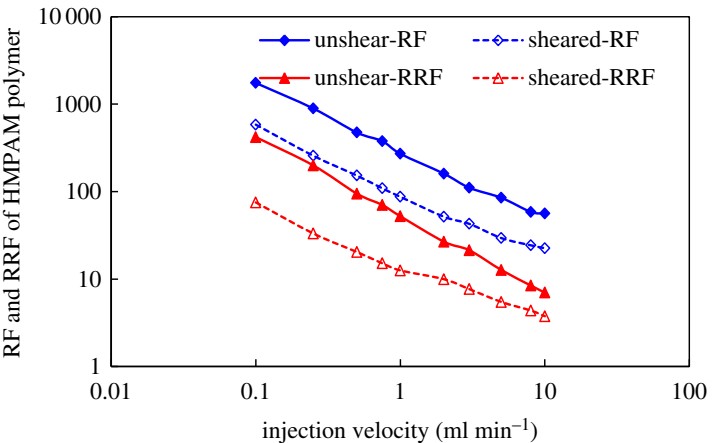

**Figure 13.** RF and RRF of HMPAM polymer at different injection velocities.

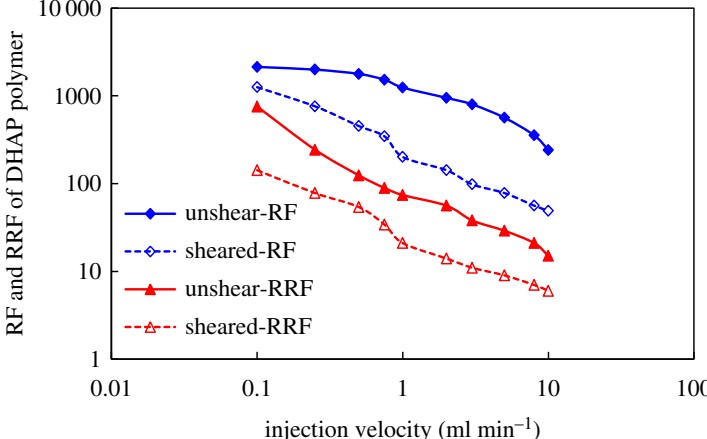

**Figure 14.** RF and RRF of DHAP polymer at different injection velocities.

molecular chain, reduces the resistance in the solution and reduces its adsorption and retention ability in porous media, weakening its fluidity control ability and making it easier to achieve a dynamic equilibrium state. In addition, pre-shearing polymer solution can ensure the injectability of the solution, especially DHAP with cluster aggregation behaviour, when the hydrophobically associated polymer solution with stronger viscoelasticity is used for oil displacement.

### 3.3.3. Percolation characteristics of polymers in porous media

RF and RRF were established in porous media using HMPAM and DHAP polymers before and after mechanical shearing. The results are shown in figures 13 and 14.

The resistance coefficients and residual resistance coefficients of the two polymers are basically the same as the injection rate is altered, exhibiting a reduction as the injection rate increased. In addition, some differences in the downward trend are noted. This finding is attributed to the fact that both are hydrophobic associative polymers. The structural viscosity is dominant in the formation of association. As the flow resistance of the solution increases, the association with the polymer molecules adsorbed on the rock surface forms multilayer adsorption, which reduces the pore size of porous media and enhances the adsorption of molecules and porous media in polymer solution. Regarding intermolecular forces, intermolecular association is a reversible physical cross-linking. As shear stress increases, the association decreases. In addition, the macromolecular aggregates and adsorption layers are dismantled, resulting in the constant reduction drag coefficient and residual drag coefficient. The spatial structure of aggregation behaviour formed by DHAP is stronger, so it is subjected to shear. In addition, the degree of association weakening is relatively low.

After mechanical shearing, polymer molecular aggregates are more easily disassembled to form small molecular clusters, which are washed out by water, thus reducing the adsorption retention of solution in

porous media and resulting in a reduction in residual resistance coefficient. DHAP with clustered aggregation characteristics exhibits strong shear resistance due to its strong spatial structure, which makes the RF and RRF established in porous media stronger than HMPAM under the same conditions.

# 4. Conclusion

(1) As the solution concentration increases, the polymer aggregates into aggregates with different morphologies under association. At the critical association concentration, DHAP polymer forms 'cluster' aggregates, and HMAP polymer forms 'chain bundle' aggregates.

(2) Mechanical shear can destroy the molecular chains of polymers, resulting in a significant decrease in intrinsic viscosity. However, the effect on hydrophobic microregions is almost negligible. Fractured polymer molecular chains can restore part of the network structure under association. The intermolecular interaction force of DHAP is stronger than that of HMPAM. After shear failure, DHAP still exhibits a better network structure. Its apparent viscosity decrease is relatively small, while its static adsorption decreases slightly.

(3) When polymer can build stronger RF and RRF in porous media, practical problems of poor injection performance will occur. When polymer solution is subjected to mechanical shear, its conductivity in porous media can be improved while its performance is reduced. Therefore, proper pre-shear is beneficial to polymer injection with strong solution performance.

Data accessibility. This article has no additional data.

Authors' contributions. L.S., S.Z., Z.Y. and J.Z. designed this study. S.Z. prepared all samples for analysis. S.Z., X.X. and W.Z. collected and analysed the data. L.S. and J.Z. interpreted the results. S.Z. wrote the manuscript. All authors gave final approval for publication.

Competing interests. The authors declare no competing interests.

Funding. Financial support was from the '13th Five-Year' National Science and Technology Major Project 25 task 3 'offshore oilfield chemical flooding technology' (no. 2016ZX05025-003).

Acknowledgements. We thank instructional support State Key Laboratory of Oil & Gas Reservoir and Exploitation Engineering, Southwest Petroleum University.

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
