## [Reviewer comments · Royal Society Open Science]

Review History

RSOS-191270.R0 (Original submission)

Review form: Reviewer 1

Is the manuscript scientifically sound in its present form?

No

Are the interpretations and conclusions justified by the results?

No

Is the language acceptable?

No

Do you have any ethical concerns with this paper?

No

Have you any concerns about statistical analyses in this paper?

No

Recommendation?

Reject

Comments to the Author(s)

This manuscript describes the aggregation behaviour characteristics of hydrophobically modified partially hydrolysed polyacrylamide and branched hydrophobically associated polymer; reports the influence of shearing on the viscosity enhancement, aggregation behaviour, microstructure and static adsorption properties of polymer solutions and describes the percolation characteristics of polymers before and after mechanical shearing in porous media at different velocities. It has not been written carefully and not checked by the authors before submission. The organization of the manuscript is not good. The manuscript is not suitable for publication in the present form; it may be resubmitted after careful revision.

Some of my opinions and comments about this manuscript are as follows:

- 1) Page 2, lines 19-28: Please use past tenses in these sentences.
- 2) Page 2, lines 22 and 23: Abbreviations "DHAP" and "HMPAM" should be defined at first mention.
- 3) Page 3, line 9: After this sentence, I suggest the authors that the following sentence and reference should be given.
 "The use of polymers to control the stability and flocculation behavior of some dispersions has great technological importance [3]".
 • [3] Journal of Applied Polymer Science, 109 (2008) 1850-1860.
- 4) Page 4, lines 18-26: These paragraphs should be re-written.
- 5) Page 4, lines 33-39 and 54-56 and page 5, lines 3-14: These sentences should be revised.
- 6) Page 5, lines 22-42: Please use subsections in section 2.2.1.
- 7) Page 5, lines 54-56 and page 6, lines 3 and 13-18: These sentences should be re-written. Do not use imperative sentences.
- 8) Page 6, lines 28-33, Table 1:
 - "Tab." should be replaced by "Table".
 - "Tab. 1" should be given in Supplementary Materials.
 - The title of Tab. 1 should be revised.
- 9) Page 6, lines 24-47: Please use subsections in section 2.2.4.
- 10) Page 6, line 42: Please check the section number given as "1.2.5 percolation characteristic experiment".
- 11) Page 6, lines 54-56 and page 7, line 3: These sentences should be revised.
- 12) Page 7, Fig. 4: Please check the spelling of "ovem".
- 13) Page 8, Figs. 5 and 6: The names of x-axes in Figs. 5 and 6 should be given as "Wavelength", not "wave length" or "Wave length".
- 14) Page 10, lines 16-18, Table 2:
 - "Tab." should be replaced by "Table".
 - Please indicate the unit of apparent viscosity.

- 15) Page 10, line 49: Please check the section number given as “(see 1.1 materials and equipment)”.
- 16) Page 14, Figs. 12 and 13: In the x-axes of Figs. 12 and 13, please check the spelling of “volum”.
- 17) Pages 17-19, “References” section:
- There is no coherence in the use of capital and small letters during the spelling of article titles. See references [1] and [6].
 - In reference [11], please indicate the volume and page numbers of article.
- 18) Pages 20-34:
- What are the titles of these tables?
 - Furthermore, they should be given in Supplementary Materials.
- 19) Pages 35 and 36:
- What are the titles of these tables?
 - These tables should be re-organized.
 - Please use subscript for “1” and “3” in “I1”, “I3” and “I1/I3”.
- 20) Page 37:
- What are the titles of these tables?
 - These tables should be re-organized.
 - Although the concentration values are same, why did the authors give different viscosity values in these tables?
- 21) Pages 38-49:
- What are the titles of these tables?
 - They should be given in Supplementary Materials.
- 22) Pages 56 and 57:
- What is the title of this table?
 - Check the data given in the 1st column. I could not see them in this table.
 - It should be given in Supplementary Materials.
- 23) Pages 58 and 59:
- What are the titles of these tables?
 - These tables should be re-organized.
 - These tables should be combined.
- 24) Page 60: Please indicate the title of this table.
- 25) Pages 61-66:
- What are the titles of these tables?
 - These tables should be re-organized.

Review form: Reviewer 2

Is the manuscript scientifically sound in its present form?

Yes

Are the interpretations and conclusions justified by the results?

Yes

Is the language acceptable?

No

Do you have any ethical concerns with this paper?

No

Have you any concerns about statistical analyses in this paper?

No

Recommendation?

Accept with minor revision (please list in comments)

Comments to the Author(s)

This paper lays the groundwork for the application of polymer flooding technology in oilfield, studies the performance of polymer solutions with different aggregation behavior and their percolation characteristics in porous media. The correlation between polymer micro-aggregation behavior and macro-percolation is established to guide polymer synthesis according to the actual needs of oilfields. The whole content, structure and basic data of the paper are relatively perfect, which has certain application guiding value. Here are some suggestions and questions:

- 1) Whether mechanical shear action and mechanical degradation are meaningful, and whether the effect of mechanical shear action is consistent with that of polymer. Mechanical degradation is used more frequently in terminology.
- 2) Figures 8 and 9, which have two patterns, are best described in subheadings, such as (a)..... and (b) .
- 3) For the first time in the abstract, HMPAM and DHAP should be introduced in full and then coded.
- 4) The format of papers must conform to the periodical norms
- 5) The introduction of synthetic drugs in this article is not perfect, including name, manufacturer and so on, so as to facilitate the experimental repeatability of other scholars.

Review form: Reviewer 3

Is the manuscript scientifically sound in its present form?

Yes

Are the interpretations and conclusions justified by the results?

No

Is the language acceptable?

Yes

Do you have any ethical concerns with this paper?

No

Have you any concerns about statistical analyses in this paper?

No

Recommendation?

Major revision is needed (please make suggestions in comments)

Comments to the Author(s)

Overall, there are a number of serious flaws and the authors should address the following issues before it might possibly be published.

P2 Abstract "Mechanical shear pretreatment effectively improves the injection conductivity of the polymer."

Please explain "injection conductivity".

P3 "Although the bulk viscosity is irreversibly damaged, the structural viscosity is reconnected to form part of the spatial structure under the association effect, and the solution viscosity is restored."

This sentence is confusing. Please rewrite this sentence to clarify what the authors mean.

P4 Experiment

Past tense should be used in describing the experiment procedure.

P4 "Experimental drugs": change to "Chemicals".

P4 "Experimental equipment": change to "Instruments".

P4 "Experimental process device for core displacement". Please explain what it is.

P4 "the other is DHAP..." Please include the full name of DHAP here.

P4 "4.8 mg ammonium persulfate are added for 6-10 hours": change to "4.8 mg ammonium persulfate are added and then stirred for another 6-10 hours".

P4 "Then, 20.16 g and 8.09 g of two methylene sulfoxide are added..." By methylene sulfoxide, I assume the authors mean dimethyl sulfoxide.

P4 Fig.2 The molecular structure of the HMPAM

Please redo this scheme (using ChemDraw, etc.) to make it better.

P5 "0.26 g solvent urea..." What is solvent urea?

P9 3.1.2 Analysis of Microstructure Characteristics by AFM

The AFM sample was prepared with evaporating the solvent with nitrogen flow. However, the structure of the polymer aggregates are very likely to change during the solvent loss process. Therefore the images can not be used as morphology representation for polymers in solution. I suggest the authors remove Fig. 8 and whole 3.1.2 discussion.

P10 Tab.2 The effect of shear strength on the intrinsic viscosity and Huggins constant of polymer
What is the unit for apparent viscosity in this table, Pa*s or Poise?

P10 "Compared with DHAP (molecular weight 600)"

Please correct the number.

P11 Tab.3 Effect of different shear strengths on the number of hydrophobic microregions in polymer solution

Can the authors please elaborate on the Shear mode?

P11 "As shear stress increases, the I_e/I_m of 2000 mg/L HMPAM solution..."

Please define I_e and I_m here.

P11-12 "Based on the effect of shear action on the hydrophobic microregion, we think the shear mainly destroys the polymer molecular chains."

It is weird. It was likely not molecular chains but intermolecular packing that broke. You can never observe molecular chain breaking with SEM.

P12 "The shear action causes the breakage of some polymer chains and aggregates, ..."

The experiment was static adsorption, so why authors believe the polymers were sheared?

P19-P67

This part should be rewritten into some form of Supporting Information.

Decision letter (RSOS-191270.R0)

23-Sep-2019

Dear Dr Zhu:

Title: The Seepage Flow Characteristics of Hydrophobically Associated Polymers with Different Aggregation Behaviours in Porous Media

Manuscript ID: RSOS-191270

The editor assigned to your manuscript has now received comments from reviewers. We would like you to revise your paper in accordance with the referee and Subject Editor suggestions which can be found below (not including confidential reports to the Editor). Please note this decision does not guarantee eventual acceptance.

Please submit your revised paper before 16-Oct-2019. Please note that the revision deadline will expire at 00.00am on this date. If we do not hear from you within this time then it will be assumed that the paper has been withdrawn. In exceptional circumstances, extensions may be possible if agreed with the Editorial Office in advance. We do not allow multiple rounds of revision so we urge you to make every effort to fully address all of the comments at this stage. If deemed necessary by the Editors, your manuscript will be sent back to one or more of the original reviewers for assessment. If the original reviewers are not available we may invite new reviewers.

Royal Society of Chemistry
Thomas Graham House
Science Park, Milton Road
Cambridge, CB4 0WF

Royal Society Open Science - Chemistry Editorial Office

RSC Subject Editor:
Comments to the Author:
(There are no comments.)

RSC Associate Editor:
Comments to the Author:
(There are no comments.)

Reviewers' Comments to Author:
Reviewer: 1

Comments to the Author(s)

This manuscript describes the aggregation behaviour characteristics of hydrophobically modified partially hydrolysed polyacrylamide and branched hydrophobically associated polymer; reports the influence of shearing on the viscosity enhancement, aggregation behaviour, microstructure and static adsorption properties of polymer solutions and describes the percolation characteristics of polymers before and after mechanical shearing in porous media at different velocities. It has not been written carefully and not checked by the authors before submission. The organization of the manuscript is not good. The manuscript is not suitable for publication in the present form; it may be resubmitted after careful revision.

Some of my opinions and comments about this manuscript are as follows:

- 1) Page 2, lines 19-28: Please use past tenses in these sentences.
- 2) Page 2, lines 22 and 23: Abbreviations "DHAP" and "HMPAM" should be defined at first mention.
- 3) Page 3, line 9: After this sentence, I suggest the authors that the following sentence and reference should be given.
"The use of polymers to control the stability and flocculation behavior of some dispersions has great technological importance [3]".
 - [3] Journal of Applied Polymer Science, 109 (2008) 1850-1860.
- 4) Page 4, lines 18-26: These paragraphs should be re-written.
- 5) Page 4, lines 33-39 and 54-56 and page 5, lines 3-14: These sentences should be revised.
- 6) Page 5, lines 22-42: Please use subsections in section 2.2.1.
- 7) Page 5, lines 54-56 and page 6, lines 3 and 13-18: These sentences should be re-written. Do not use imperative sentences.
- 8) Page 6, lines 28-33, Table 1:
 - "Tab." should be replaced by "Table".
 - "Tab. 1" should be given in Supplementary Materials.
 - The title of Tab. 1 should be revised.

- 9) Page 6, lines 24-47: Please use subsections in section 2.2.4.
- 10) Page 6, line 42: Please check the section number given as “1.2.5 percolation characteristic experiment”.
- 11) Page 6, lines 54-56 and page 7, line 3: These sentences should be revised.
- 12) Page 7, Fig. 4: Please check the spelling of “ovem”.
- 13) Page 8, Figs. 5 and 6: The names of x-axes in Figs. 5 and 6 should be given as “Wavelength”, not “wave length” or “Wave length”.
- 14) Page 10, lines 16-18, Table 2:
- “Tab.” should be replaced by “Table”.
 - Please indicate the unit of apparent viscosity.
- 15) Page 10, line 49: Please check the section number given as “(see 1.1 materials and equipment)”.
- 16) Page 14, Figs. 12 and 13: In the x-axes of Figs. 12 and 13, please check the spelling of “volum”.
- 17) Pages 17-19, “References” section:
- There is no coherence in the use of capital and small letters during the spelling of article titles. See references [1] and [6].
 - In reference [11], please indicate the volume and page numbers of article.
- 18) Pages 20-34:
- What are the titles of these tables?
 - Furthermore, they should be given in Supplementary Materials.
- 19) Pages 35 and 36:
- What are the titles of these tables?
 - These tables should be re-organized.
 - Please use subscript for “1” and “3” in “I1”, “I3” and “I1/I3”.
- 20) Page 37:
- What are the titles of these tables?
 - These tables should be re-organized.
 - Although the concentration values are same, why did the authors give different viscosity values in these tables?
- 21) Pages 38-49:
- What are the titles of these tables?
 - They should be given in Supplementary Materials.
- 22) Pages 56 and 57:
- What is the title of this table?
 - Check the data given in the 1st column. I could not see them in this table.
 - It should be given in Supplementary Materials.
- 23) Pages 58 and 59:
- What are the titles of these tables?
 - These tables should be re-organized.
 - These tables should be combined.

24) Page 60: Please indicate the title of this table.

25) Pages 61-66:

- What are the titles of these tables?
- These tables should be re-organized.

Reviewer: 2

Comments to the Author(s)

This paper lays the groundwork for the application of polymer flooding technology in oilfield, studies the performance of polymer solutions with different aggregation behavior and their percolation characteristics in porous media. The correlation between polymer micro-aggregation behavior and macro-percolation is established to guide polymer synthesis according to the actual needs of oilfields. The whole content, structure and basic data of the paper are relatively perfect, which has certain application guiding value. Here are some suggestions and questions:

- 1) Whether mechanical shear action and mechanical degradation are meaningful, and whether the effect of mechanical shear action is consistent with that of polymer. Mechanical degradation is used more frequently in terminology.
- 2) Figures 8 and 9, which have two patterns, are best described in subheadings, such as (a)..... and (b) .
- 3) For the first time in the abstract, HMPAM and DHAP should be introduced in full and then coded.
- 4) The format of papers must conform to the periodical norms
- 5) The introduction of synthetic drugs in this article is not perfect, including name, manufacturer and so on, so as to facilitate the experimental repeatability of other scholars.

Reviewer: 3

Comments to the Author(s)

Overall, there are a number of serious flaws and the authors should address the following issues before it might possibly be published.

P2 Abstract "Mechanical shear pretreatment effectively improves the injection conductivity of the polymer."

Please explain "injection conductivity".

P3 "Although the bulk viscosity is irreversibly damaged, the structural viscosity is reconnected to form part of the spatial structure under the association effect, and the solution viscosity is restored."

This sentence is confusing. Please rewrite this sentence to clarify what the authors mean.

P4 Experiment

Past tense should be used in describing the experiment procedure.

P4 "Experimental drugs": change to "Chemicals".

P4 "Experimental equipment:": change to "Instruments".

P4 "Experimental process device for core displacement". Please explain what it is.

P4 "the other is DHAP..." Please include the full name of DHAP here.

P4 "4.8 mg ammonium persulfate are added for 6-10 hours": change to "4.8 mg ammonium persulfate are added and then stirred for another 6-10 hours".

P4 "Then, 20.16 g and 8.09 g of two methylene sulfoxide are added..." By methylene sulfoxide, I assume the authors mean dimethyl sulfoxide.

P4 Fig.2 The molecular structure of the HMPAM
Please redo this scheme (using ChemDraw, etc.) to make it better.

P5 "0.26 g solvent urea..." What is solvent urea?

P9 3.1.2 Analysis of Microstructure Characteristics by AFM
The AFM sample was prepared with evaporating the solvent with nitrogen flow. However, the structure of the polymer aggregates are very likely to change during the solvent loss process. Therefore the images can not be used as morphology representation for polymers in solution. I suggest the authors remove Fig. 8 and whole 3.1.2 discussion.

P10 Tab.2 The effect of shear strength on the intrinsic viscosity and Huggins constant of polymer
What is the unit for apparent viscosity in this table, Pa*s or Poise?

P10 "Compared with DHAP (molecular weight 600)"
Please correct the number.

P11 Tab.3 Effect of different shear strengths on the number of hydrophobic microregions in polymer solution
Can the authors please elaborate on the Shear mode?

P11 "As shear stress increases, the I_e/I_m of 2000 mg/L HMPAM solution..."
Please define I_e and I_m here.

P11-12 "Based on the effect of shear action on the hydrophobic microregion, we think the shear mainly destroys the polymer molecular chains."
It is weird. It was likely not molecular chains but intermolecular packing that broke. You can never observe molecular chain breaking with SEM.

P12 "The shear action causes the breakage of some polymer chains and aggregates, ..."
The experiment was static adsorption, so why authors believe the polymers were sheared?

P19-P67
This part should be rewritten into some form of Supporting Information.

Author's Response to Decision Letter for (RSOS-191270.R0)

See Appendix A.

RSOS-191270.R1 (Revision)

Review form: Reviewer 3

Is the manuscript scientifically sound in its present form?

Yes

Are the interpretations and conclusions justified by the results?

Yes

Is the language acceptable?

Yes

Do you have any ethical concerns with this paper?

No

Have you any concerns about statistical analyses in this paper?

No

Recommendation?

Accept with minor revision (please list in comments)

Comments to the Author(s)

The authors addressed most of my questions and I appreciate it. A minor request:

Original question: 9) P11 "As shear stress increases, the I_e/I_m of 2000 mg/L HMPAM solution...". Please define I_e and I_m here.

Response:

Authors' response: According to expert opinion, relevant explanations were added after the table, I_e -Maximum Excipient Association Peak; I_m -Maximum Emission Peak of Monomer in Emission Spectrum.

My comment: Please elaborate on Maximum Excipient Association Peak.

Decision letter (RSOS-191270.R1)

05-Nov-2019

Dear Dr zhu:

Title: The Seepage Flow Characteristics of Hydrophobically Associated Polymers with Different Aggregation Behaviours in Porous Media

Manuscript ID: RSOS-191270.R1

Thank you for submitting the above manuscript to Royal Society Open Science. On behalf of the Editors and the Royal Society of Chemistry, I am pleased to inform you that your manuscript will be accepted for publication in Royal Society Open Science subject to minor revision in accordance with the referee suggestions. Please find the reviewers' comments at the end of this email.

The reviewers and handling editors have recommended publication, but also suggest some minor revisions to your manuscript. Therefore, I invite you to respond to the comments and revise your manuscript.

Because the schedule for publication is very tight, it is a condition of publication that you submit the revised version of your manuscript before 14-Nov-2019. Please note that the revision deadline will expire at 00.00am on this date. If you do not think you will be able to meet this date please let me know immediately.

To revise your manuscript, log into <https://mc.manuscriptcentral.com/rsos> and enter your

Author Centre, where you will find your manuscript title listed under "Manuscripts with Decisions". Under "Actions," click on "Create a Revision." You will be unable to make your revisions on the originally submitted version of the manuscript. Instead, revise your manuscript and upload a new version through your Author Centre.

Best wishes,
Dr Laura Smith
Publishing Editor, Journals

RSC Associate Editor:
Comments to the Author:
(There are no comments.)

RSC Subject Editor:
Comments to the Author:
(There are no comments.)

Reviewer comments to Author:
Reviewer: 3

Comments to the Author(s)
The authors addressed most of my questions and I appreciate it. A minor request:

Original question: 9) P11 "As shear stress increases, the I_e/I_m of 2000 mg/L HMPAM solution...". Please define I_e and I_m here.

Response:

Authors' response: According to expert opinion, relevant explanations were added after the table, I_e -Maximum Excipient Association Peak; I_m -Maximum Emission Peak of Monomer in Emission Spectrum.

My comment: Please elaborate on Maximum Excipient Association Peak.

Author's Response to Decision Letter for (RSOS-191270.R1)

See Appendix B.

RSOS-191270.R2 (Revision)

Review form: Reviewer 3

Is the manuscript scientifically sound in its present form?

Yes

Are the interpretations and conclusions justified by the results?

Yes

Is the language acceptable?

Yes

Do you have any ethical concerns with this paper?

No

Have you any concerns about statistical analyses in this paper?

No

Recommendation?

Accept as is

Comments to the Author(s)

Looks OK.

Decision letter (RSOS-191270.R2)

02-Dec-2019

Dear Dr zhu:

Title: The Seepage Flow Characteristics of Hydrophobically Associated Polymers with Different Aggregation Behaviours in Porous Media
Manuscript ID: RSOS-191270.R2

It is a pleasure to accept your manuscript in its current form for publication in Royal Society Open Science. The chemistry content of Royal Society Open Science is published in collaboration with the Royal Society of Chemistry.

RSC Associate Editor:
Comments to the Author:
(There are no comments.)

RSC Subject Editor:
Comments to the Author:
(There are no comments.)

Reviewer(s)' Comments to Author:
Reviewer: 3

Comments to the Author(s)
Looks OK.

Appendix A

Dear Editor,

Thank you very much for your attention and the referees' valuable comments on our paper. We have revised the manuscript according to reviewers' comments. Enclosed please find the revised manuscript, responses to the referees as well as a list of changes. We sincerely hope this manuscript will be finally acceptable to be published on **Royal Society Open Science**. We look forward to hearing from you soon.

Best regards

Dr. zhu

Reviewers' Comments to Author:

Reviewer: 1

Comments to the Author(s): This manuscript describes the aggregation behaviour characteristics of hydrophobically modified partially hydrolysed polyacrylamide and branched hydrophobically associated polymer; reports the influence of shearing on the viscosity enhancement, aggregation behaviour, microstructure and static adsorption properties of polymer solutions and describes the percolation characteristics of polymers before and after mechanical shearing in porous media at different velocities. It has not been written carefully and not checked by the authors before submission. The organization of the manuscript is not good. The manuscript is not suitable for publication in the present form; it may be resubmitted after careful revision.

Some of my opinions and comments about this manuscript are as follows:

1) Page 2, lines 19-28: Please use past tenses in these sentences.

Response:

According to experts' opinions, the contents of 19-28 are written in the past tense.

The revises are as follows:

“The results were showed that mechanical shearing did not affect hydrophobic microzones. Polymers can re-associate to restore part of the network structure, thereby improving shear resistance (dendritic hydrophobically associating polymers > hydrophobically modified partially hydrolysed polyacrylamide). Polymers with "cluster" aggregation behaviour enhanced solution performance, enabling them to establish higher resistance coefficient (RF) and residual resistance factor (RRF) in porous media but also bringing about injection difficulties. Increasing the injection rate would increase the injection pressure, but the established RF and RRF showed a downward trend. Mechanical shear pretreatment effectively improved the injectivity of the polymer.”

2) Page 2, lines 22 and 23: Abbreviations “DHAP” and “HMPAM” should be defined at first mention.

Response:

The revises are as follows:

“DHAP” change to “dendritic hydrophobically associating polymers”

“HMPAM” change to “hydrophobically modified partially hydrolysed polyacrylamide”.

3) Page 3, line 9: After this sentence, I suggest the authors that the following sentence

and reference should be given. “The use of polymers to control the stability and flocculation behavior of some dispersions has great technological importance [3]”.

• [3] *Journal of Applied Polymer Science*, 109 (2008) 1850-1860.

Response:

According to the expert's suggestion, language and references were added.

4) Page 4, lines 18-26: These paragraphs should be re-written.

Response:

Revisions were made on the basis of expert advice.

The revises are as follows:

Chemicals: sodium chloride, calcium chloride, magnesium chloride, acrylamide (AM), acrylic acid (AA), two methyl allyl pair of sixteen alkyl benzyl ammonium chloride, two methyl allyl -N- alkyl ammonium chloride, maleic anhydride, macromolecular skeleton monomer.

Experimental brine: The experimental brine with 3000.00 mg/L Na⁺ and 300.00 mg/L Ca²⁺/Mg²⁺.

Instruments: ① Fluorescence spectrometer (Cary Eclipse); ② The Ubbelohde viscometer; ③ Scanning Electron Microscope (SEM); ④ Experimental process device for core displacement; ⑤ Ultraviolet spectrophotometer; ⑥ Nanoscope IIIa microscope.

5) Page 4, lines 33-39 and 54-56 and page 5, lines 3-14: These sentences should be revised.

Response:

Revisions were made on the basis of expert advice.

The revises are as follows:

The first step involves weighing 12 g of acrylamide (AM), 3 g of acrylic acid (AA), 0.369 g of two methyl allyl pair of sixteen alkyl benzyl ammonium chloride, 35 g of distilled water, respectively, to a beaker, stirring and dissolving evenly. Then, the pH is adjusted to 5-6 by using sodium carbonate. After heating for 45 minutes in a water bath at a temperature of 45 degrees, adding 4.8 mg initiator sodium bisulfite and 4.8 mg ammonium persulfate, and reacting for 6-10 hours, the hydrophobically modified partially hydrolysed polyacrylamide polymer could be obtained.

The molecular structure is presented in Fig.2. Synthesis steps include weigh 12 grams of acrylamide (AM), 3 grams of acrylic acid (AA), 0.369 grams of two methyl allyl pair of sixteen alkyl benzyl ammonium chloride, 35 grams of distilled water; add to the beaker, stir and dissolve evenly. The pH is adjusted to 5-6 using sodium carbonate. After heating for 45 minutes in the water bath at a temperature of 45 degrees, 4.8 mg initiator sodium bisulfite and 4.8 mg ammonium persulfate are added and then stirred for another 6-10 hours, and the hydrophobically modified partially hydrolysed polyacrylamide polymer was obtained.

6) Page 5, lines 22-42: Please use subsections in section 2.2.1.

Response:

According to expert opinions, the articles are classified into sections.

7) Page 5, lines 54-56 and page 6, lines 3 and 13-18: These sentences should be

re-written. Do not use imperative sentences.

Response:

Revisions were made on the basis of expert advice.

The revises are as follows:

The synthetic steps are as follows: first, the two generations of polyamide amine tree-like macromolecular skeleton monomer, 2.619 g and maleic anhydride, 1.437 g were added to different beakers, and then, 20.16 g and 8.09 g of two methylene sulfoxides were added to evenly stir the solution. According to the mole ratio of 1:8, the two methylene sulfoxide solution of skeleton monomers is slowly dripped into the two methylene sulfoxide solution of maleic anhydride, nitrogen is added in the process of dripping, and the system is put into an ice salt bath. After adding the skeleton monomer solution and stirring until the temperature of the reaction liquid system no longer rises (temperature is no more than 50 degrees), the functional skeleton monomer (C₉₄H₁₄₄O₃₆N₂₆) is obtained by repeated precipitation with chloroform. The next step involves weighing 20 g acrylamide (AM), 5 g acrylic acid (AA), 0.025 g functionalized skeleton monomer, 0.8 g two methyl allyl -N- alkyl ammonium chloride, 74 g distilled water, and pouring into a beaker to stir and dissolve. Sodium hydroxide is used to adjust pH to 5-6 and the temperature set to 35 °C to preheat for 30 minutes. Adding 0.26 g solvent urea, 0.26 g sodium sulfate, 0.052 g chain transfer agent sodium methoxide, 4.8 mg oxidation reduction initiator sodium pisolite and 4.8 mg ammonium persulfate, and reacting for 8 hours results in the DHAP polymer product.

8) Page 6, lines 28-33, Table 1: • “Tab.” should be replaced by “Table”. • “Tab. 1” should be given in Supplementary Materials. • The title of Tab. 1 should be revised.

Response:

According to the experts'opinions, the relevant contents were revised and "tab." was changed to "table".

9) Page 6, lines 24-47: Please use subsections in section 2.2.4.

Response:

According to expert opinions, the articles are classified into sections.

10) Page 6, line 42: Please check the section number given as “1.2.5 percolation characteristic experiment”.

Response:

Thank you for the mistake pointed out by the experts. The correct number is 2.2.8.

11) Page 6, lines 54-56 and page 7, line 3: These sentences should be revised.

Response:

According to the expert's opinion, the amendments are as follows.

“The experimental contents are as follows: Two polymer solutions with concentration of 2000 mg/L were prepared. A part of the solution was used to simulate mechanical shear. The simulation condition was 3500 rpm shear for 20 seconds. The prepared polymer solution was injected into porous media at a certain injection rate of 0.5 ml/min, 1 ml/min and 3 ml/min, respectively.”

12) Page 7, Fig. 4: Please check the spelling of “ovem”.

Response:

This mistake have been revised.

Figure 4.

13) Page 8, Figs. 5 and 6: The names of x-axes in Figs. 5 and 6 should be given as “Wavelength”, not “wave length” or “Wave length”.

Response:

Revisions were made in accordance with the recommendations of experts.

Figure 5

Figure 6.

14) Page 10, lines 16-18, Table 2: • “Tab.” should be replaced by “Table”. • Please indicate the unit of apparent viscosity.

Response:

The unit of apparent viscosity is $\text{mPa}\cdot\text{s}$

15) Page 10, line 49: Please check the section number given as “(see 1.1 materials and equipment)”.

Response:

The correct number is 2.1 materials and equipment

16) Page 14, Figs. 12 and 13: In the x-axes of Figs. 12 and 13, please check the spelling of “volum”.

Response:

The problem in the diagram was revised. “Volum” change to “Volume”.

Figure 12.

Figure 13.

17) Pages 17-19, “References” section:

• There is no coherence in the use of capital and small letters during the spelling of article titles. See references [1] and [6].

• In reference [11], please indicate the volume and page numbers of article.

Response:

The format was revised according to the periodical requirements.

18) Pages 20-34:

• What are the titles of these tables?

• Furthermore, they should be given in Supplementary Materials.

Response:

The original data in Figures 5 and 6.

19) Pages 35 and 36:

- **What are the titles of these tables?**
- **These tables should be re-organized.**
- **Please use subscript for “1” and “3” in “I1”, “I3” and “I1/I3”.**

Response:

The original data in Figure 7.

20) Page 37:

- **What are the titles of these tables?**
- **These tables should be re-organized.**
- **Although the concentration values are same, why did the authors give different viscosity values in these tables?**

Response:

The original data in Figure 10.

21) Pages 38-49:

- **What are the titles of these tables?**
- **They should be given in Supplementary Materials.**

Response:

The original data in Figure 11.

22) Pages 56 and 57:

- **What is the title of this table?**
- **Check the data given in the 1st column. I could not see them in this table.**
- **It should be given in Supplementary Materials.**

The original data in Figure 12

23) Pages 58 and 59:

- **What are the titles of these tables?**
- **These tables should be re-organized.**
- **These tables should be combined.**

24) Page 60: Please indicate the title of this table.

25) Pages 61-66:

- **What are the titles of these tables?**
- **These tables should be re-organized.**

Response:

The original data in Figures 13 and 14.

In this part, the author uses Excel to organize the original data of each table and graph. In the process of transforming into PDF, there is a relatively unclear presentation. In fact, this part of the data is clearly labeled in the process of uploading.

Reviewer: 2

Comments to the Author(s): This paper lays the groundwork for the application of polymer flooding technology in oilfield, studies the performance of polymer solutions with different aggregation behavior and their percolation characteristics in porous media. The correlation between polymer micro-aggregation behavior and macro-percolation is established to guide polymer synthesis according to the actual needs of oilfields. The whole content, structure and basic data of the paper are relatively perfect, which has certain application guiding value. Here are some suggestions and questions:

1) Whether mechanical shear action and mechanical degradation are meaningful, and whether the effect of mechanical shear action is consistent with that of polymer. Mechanical degradation is used more frequently in terminology.

Response:

Thank you very much for your questions. The physical effect of mechanical shearing is discussed in this paper, but it is not discussed from the degradation of polymers. So the method used in this paper is mechanical shearing.

2) Figures 8 and 9, which have two patterns, are best described in subheadings, such as (a)..... and (b)

Response:

According to the expert's opinions, (a) (b) classified description was made.

3) For the first time in the abstract, HMPAM and DHAP should be introduced in full and then coded.

Response:

Revised on the basis of expert advice.

“DHAP” change to “dendritic hydrophobically associating polymers”

“HMPAM” change to “hydrophobically modified partially hydrolysed polyacrylamide”.

4) The format of papers must conform to the periodical norms

Response:

The format of the article was revised, including figures, tables and references.

5) The introduction of synthetic drugs in this article is not perfect, including name, manufacturer and so on, so as to facilitate the experimental repeatability of other scholars.

Response:

Chemicals are analytical Reagent (AR), and general chemical products are sold by merchants.

Reviewer: 3

Comments to the Author(s). Overall, there are a number of serious flaws and the authors should address the following issues before it might possibly be published.

1) P2 Abstract “Mechanical shear pretreatment effectively improves the injection

conductivity of the polymer.” Please explain “injection conductivity”.

Response:

What is described here is to improve the injectability of polymers. So change “injection conductivity” to “injectability”.

2) P3 “Although the bulk viscosity is irreversibly damaged, the structural viscosity is reconnected to form part of the spatial structure under the association effect, and the solution viscosity is restored.” This sentence is confusing. Please rewrite this sentence to clarify what the authors mean.

Response:

changes to “Although bulk viscosity is irreversibly destroyed by mechanical shear, hydrophobic association can restore part of the structural viscosity, thus restoring the solution viscosity.”

3) P4 Experiment, Past tense should be used in describing the experiment procedure.

P4 “Experimental drugs”: change to “Chemicals”.

P4 “Experimental equipment”: change to “Instruments”.

P4 “Experimental process device for core displacement”. Please explain what it is.

P4 “the other is DHAP...” Please include the full name of DHAP here.

P4 “4.8 mg ammonium persulfate are added for 6-10 hours”: change to “4.8 mg ammonium persulfate are added and then stirred for another 6-10 hours”.

P4 “Then, 20.16 g and 8.09 g of two methylene sulfoxide are added...” By methylene sulfoxide, I assume the authors mean dimethyl sulfoxide.

P4 Fig.2 The molecular structure of the HMPAM, Please redo this scheme (using ChemDraw, etc.) to make it better.

P5 “0.26 g solvent urea...” What is solvent urea?

Response:

The revises are as follows:

Chemicals: sodium chloride, calcium chloride, magnesium chloride, acrylamide (AM), acrylic acid (AA), two methyl allyl pair of sixteen alkyl benzyl ammonium chloride, two methyl allyl -N- alkyl ammonium chloride, maleic anhydride, macromolecular skeleton monomer.

Experimental brine: The experimental brine with 3000.00 mg/L Na⁺ and 300.00 mg/L Ca²⁺/Mg²⁺.

Instruments: ① Fluorescence spectrometer (Cary Eclipse); ② The Ubbelohde viscometer; ③ Scanning Electron Microscope (SEM); ④ Displacement experimental equipment; ⑤ Ultraviolet spectrophotometer; ⑥ Nanoscope IIIa microscope.

Two types of hydrophobically associated polymers ¹⁴ with different molecular structures have been studied: One is hydrophobically modified partially hydrolysed polyacrylamide (HMPAM) with a hydrophobic group content of 1.2 mol% and molecular weight of 12 million; the other is dendritic hydrophobically associating polymers (DHAP) with a hydrophobic group content of 0.6 mol% and molecular weight of 6 million.

1) Synthesis of hydrophobically modified partially hydrolysed polyacrylamide. The molecular structure is presented in Figure 2. The synthetic steps are as follows. The first step involves weighing 12 g of acrylamide (AM), 3 g of acrylic acid (AA), 0.369 g of two methyl allyl

pair of sixteen alkyl benzyl ammonium chloride, 35 g of distilled water, respectively, to a beaker, stirring and dissolving evenly. Then, the pH is adjusted to 5-6 by using sodium carbonate. After heating for 45 minutes in a water bath at a temperature of 45 degrees, adding 4.8 mg initiator sodium bisulfite and 4.8 mg ammonium persulfate, and reacting for 6-10 hours, the hydrophobically modified partially hydrolysed polyacrylamide polymer could be obtained.

2) Synthesis of branched hydrophobically associating polymers. The molecular structure is presented in Figure 3. Synthesis steps are as follows. First, the two generations of the polyamide amine tree, including 2.619 g macromolecular skeleton monomer and 1.437 g maleic anhydride, were added to different beakers. Then, 20.16 g and 8.09 g of two methylene sulfoxide are added, and the solution is evenly stirred. Using a molar ratio of 1:8, the two methylene sulfoxide solutions of skeleton monomer are slowly dripped into the two methylene sulfoxide solutions of maleic anhydride, and the nitrogen is added via a dropping process. The system is placed into an ice salt bath. After adding the skeleton monomer solution, the solution is stirred until the temperature of the reaction liquid system no longer increases (temperature is less than 50 degrees). The functional skeleton monomer ($C_{94}H_{144}O_{36}N_{26}$) is obtained by repeated precipitation with chloroform. Then, 20 g acrylamide (AM), 5 g acrylic acid (AA), 0.025 g functionalized skeleton monomer, 0.8 g two methyl allyl -N- alkyl ammonium chloride, and 74 g distilled water are weighed. The components are placed into the beaker to stir and dissolve. Sodium hydroxide is used to adjust pH to 5-6, and the solution is incubated at 35 °C to preheat for 30 minutes. Then, 0.26 g solvent urea, 0.26 g sodium sulfate, 0.052 g chain transfer agent sodium methoxide, 4.8 mg oxidation-reduction initiator sodium bisulfite and 4.8 mg ammonium persulfate are added. The reaction proceeds for 8 hours, yielding DHAP polymer.

5) P9 3.1.2 Analysis of Microstructure Characteristics by AFM. The AFM sample was prepared with evaporating the solvent with nitrogen flow. However, the structure of the polymer aggregates are very likely to change during the solvent loss process. Therefore the images can not be used as morphology representation for polymers in solution. I suggest the authors remove Fig. 8 and whole 3.1.2 discussion.

Response:

Revisions were made on the basis of comments made by experts, removed Figure 8 and whole 3.1.2 discussion.

6) P10 Tab.2 The effect of shear strength on the intrinsic viscosity and Huggins constant of polymer, What is the unit for apparent viscosity in this table, Pa*s or Poise?

Response:

The unit for apparent viscosity is mPa·s.

7) P10 “Compared with DHAP (molecular weight 600)”, Please correct the number.

Response:

Thank you for the mistakes pointed out by the experts. “weight 6 million”.

8) P11 Tab.3 Effect of different shear strengths on the number of hydrophobic microregions in polymer solution. Can the authors please elaborate on the Shear mode?

Response:

The experimental part 2.2.1 was added, and the experimental means were added. The polymer solution (200 mL) was injected into the waring blender and sheared by the waring blender at different speeds.

9) P11 “As shear stress increases, the I_e/I_m of 2000 mg/L HMPAM solution...”. Please define I_e and I_m here.

Response:

According to expert opinion, relevant explanations were added after the table, I_e -Maximum Excipient Association Peak; I_m -Maximum Emission Peak of Monomer in Emission Spectrum

10) P11-12 “Based on the effect of shear action on the hydrophobic microregion, we think the shear mainly destroys the polymer molecular chains.” It is weird. It was likely not molecular chains but intermolecular packing that broke. You can never observe molecular chain breaking with SEM.

Response:

Thank you for your query. SEM characterization is indeed the structural characteristics of molecular stacks and the damage caused by shearing. The description language is revised in this paper.

The revises are as follows:

“1) When the polymer concentration is 2000 mg/L, the microchain bundle formed by the polymer DHAP is more robust than that of HMPAM; 2) Shearing action will fracture the polymer intermolecular packing. Based on the effect of shear action on the hydrophobic microregion, we think the shear mainly destroys the polymer molecular intermolecular packing. The hydrophobic region enables fractured intermolecular packing reassembly, so they can still form the space network structure²⁴⁻²⁶. Thus, the viscosity concentration curve of polymer reveals shear thickening ability, shows stronger spatial structure and is less affected by the shear effect. Overall, the shear affects the molecular structure of chain hydrophobically associated polymer to an extent, but the intermolecular packing cleavage in the hydrophobic group under the effect of association ensures that the growth trend of solution viscosity and the molecular structure also disrupt viscosity influence.”

11) P12 “The shear action causes the breakage of some polymer chains and aggregates, ...”. The experiment was static adsorption, so why authors believe the polymers were sheared?

Response:

Thank the experts for their questions. Here is a comparison of the static adsorption characteristics of polymer solutions before and after shearing. The static adsorption results under current conditions are discussed through the analysis and understanding of the previous contents.

12) P19-P67, This part should be rewritten into some form of Supporting Information.

Response:

In this part, the author uses Excel to organize the original data of each table and graph. In the process of transforming into PDF, there is a relatively unclear presentation. In fact, this part of the data is clearly labeled in the process of uploading.

Appendix B

Dear Editor,

Thank you very much for your attention and the referees' valuable comments on our paper. We have revised the manuscript according to reviewers' comments. Enclosed please find the revised manuscript, responses to the referees as well as a list of changes.

Dr. zhu

Reviewer: 3

Comments to the Author(s) The authors addressed most of my questions and I appreciate it. A minor request: Original question: 9) P11 "As shear stress increases, the I_e/I_m of 2000 mg/L HMPAM solution...". Please define I_e and I_m here.

Response: Authors' response: According to expert opinion, relevant explanations were added after the table, I_e -Maximum Excipient Association Peak; I_m -Maximum Emission Peak of Monomer in Emission Spectrum. My comment: Please elaborate on Maximum Excipient Association Peak.

Response: When the ground state pyrene in the solution is close enough to the excited state pyrene (about 4 angstroms) and forms a surface-to-surface lamination conformation, a ground state pyrene molecule can combine with it to form excimer before the decay of excited state pyrene molecule.

Py^* is an excited state pyrene, Py is a ground state pyrene, and $(\text{Py Py})^*$ is an excimer

The fluorescence peak of $(\text{Py Py})^*$ appears in the long-wave region of the fluorescence emission spectrum, called the excipient association peak.